# Four Methods to Distinguish between Fractal Dimensions in Time Series through Recurrence Quantification Analysis

**DOI:** 10.3390/e24091314

**Published:** 2022-09-19

**Authors:** Alon Tomashin, Giuseppe Leonardi, Sebastian Wallot

**Affiliations:** 1The Gonda Multidisciplinary Brain Research Center, Bar-Ilan University, Ramat-Gan 5290002, Israel; 2Institute of Psychology, University of Economics and Human Sciences, 01-043 Warsaw, Poland; 3Institute for Sustainability Education and Psychology, Leuphana University of Lüneburg, 21335 Lüneburg, Germany; 4Department of Language and Literature, Max Planck Institute of Empirical Aesthetics, 60322 Frankfurt am Main, Germany

**Keywords:** recurrence quantification analysis, fractals, monofractals, fractal time series

## Abstract

Fractal properties in time series of human behavior and physiology are quite ubiquitous, and several methods to capture such properties have been proposed in the past decades. Fractal properties are marked by similarities in statistical characteristics over time and space, and it has been suggested that such properties can be well-captured through recurrence quantification analysis. However, no methods to capture fractal fluctuations by means of recurrence-based methods have been developed yet. The present paper takes this suggestion as a point of departure to propose and test several approaches to quantifying fractal fluctuations in synthetic and empirical time-series data using recurrence-based analysis. We show that such measures can be extracted based on recurrence plots, and contrast the different approaches in terms of their accuracy and range of applicability.

## 1. Introduction

Since Gilden et al.’s [1] seminal paper, showing the presence of 1/*f*
^α^ -fluctuations in human time estimation performance, a huge interest in the presence and meaning of fractal fluctuations in human behavior has emerged. On the one hand, fractal patterns have been found in virtually all aspects of human physiology and behavior across recent studies [2,3,4,5,6,7,8,9,10,11]. On the other hand, their meaning has been intensely discussed [12,13,14,15,16,17,18,19].

Through the same period, the development and refinement of different time-series analysis techniques gained momentum, so that fractal properties could be quantified with a variety of methods, based on the power spectrum of a time series [20], their standard deviation [21] or residual fluctuations [22]—each of which has particular advantages and downsides, as well as requirements for preprocessing [21,23]. This was of central importance, because methods that are suitable for special fractals, such as box counting, are not equally applicable to time-series data [24].

In the current paper, we want to present another way of quantifying fractal fluctuations in time-series data using recurrence quantification analysis [25,26]. Our motivation for the present work is two-fold: firstly, to extend the use of recurrence plot-based methods to capture fractal properties. This is something that recurrence plot-based analyses have not been capable of. Further, to pave the way to provide an easy-to-use tool to compare fractal dimensions of time series that are well-applicable to binary data, and in the future also to multidimensional time series using multidimensional recurrence plot methods [27]. As has been suggested elsewhere [28], fractal properties in time-series data can be well-captured by the concept of (imperfectly) recurring patterns over time, and this is—as the name implies—what recurrence quantification analysis is about. Specifically, Webber [28] encouraged researchers to explore RQA as a bridge to further understand fractal systems in various fields. However, Webber did not specify how to quantify fractal fluctuations by means of recurrence plots.

Hence, the aim of the present paper is to take this next step and propose, as well as compare, novel recurrence-based approaches that can be used to quantify fractal fluctuations. In the following, each approach is introduced, tested on synthetic data, and evaluated; in addition, a Matlab (The MathWorks, Inc., Natick, MA, USA) implementation of the approaches presented in this paper is available on GitHub: https://github.com/alontom/FARQA (Accessed on 19 July 2022, see Appendix A). Finally, we discuss the individual strengths and weaknesses of each approach and relate the results to those obtained from detrended fluctuation analysis (DFA; [22]), as DFA is one of the most widely used methods with accurate performance in capturing fractal fluctuations in time-series data [29,30,31].

## 2. Methods and Results

### 2.1. Synthetic Data

In this section, we show four new approaches to differentiate between the power-law scaling exponent (α; 1/*f*^α^) based on several RQA properties. Each evaluation approach was applied to synthetic data consisting of 1026 data points with different fractal dimensions ranging from α = −1 (antipersistent) to 2 (persistent) generated by ‘power noise’ function [32] using Matlab version 2021b (The MathWorks, Inc.). For every fractal dimension, 100 time series were generated under two conditions: idealized fractal time series and a noisy fractal time series (SNR 2:1). The noise component added was drawn from a normal distribution with 50% of the SD of the idealized fractal time series. We conducted RQA without embedding (delay and embedding parameters of 1, euclidean normalization of the phase space, and radius = 0.4) on the z-scored generated time series and utilized its properties to discriminate between signals with different 1/*f* values. As a benchmark to compare against, alongside the true predetermined α, we also subjected the data to detrended fluctuation analysis (DFA; [22]). In the next sections, we will describe each of the methods and present the results of their application. After that, we will apply the methods to empirical data of a time-estimation task. Finally, we will provide a summary of the strengths and weaknesses of each method and the intercorrelations of their results.

#### 2.1.1. Detrended Fluctuations Analysis (DFA)

First, we tested the fractal properties of the dataset by applying a detrended fluctuation analysis [22]. To do so, we used the following DFA parameters: a minimum bin size of 10, a maximum of 510, linear detrending. The results are presented in Figure 1 and show that the Hurst exponents *H* estimated via the DFA scale well with the true α-values of the time series. In the absence of random noise, DFA distinguishes scaling relations well down to antipersistent fluctuations with α = −1 (Figure 1, left panel, *R*^2^ = 0.997). When noise is added, the capacity of DFA to distinguish among antipersistent was slightly compromised (Figure 1, right panel, *R*^2^ = 0.965).

#### 2.1.2. First Approach: Estimating Scaling Using the *SD* of *%REC* over a Range of Bin Sizes (*%REC SD*)

To capture the change in fluctuations with scale, the RP was split into bins of various sizes (powers of two). In each, we calculated the recurrence rate. Then, the *SD* of all the bins of the same size was computed, and we fitted a linear line to the log–log plot of the *SD* vs. the bin sizes. Figure 2 illustrates the approach.

The rationale behind this approach is that a time series of i.i.d. white noise will yield a recurrence plot that is statistically uniformly populated by mostly isolated recurrence points, while the correlation structure of persistent fluctuations will yield a more clustered, nonuniform distribution, and will hence lead to a slower increase in *SD* compared to the white noise case (Figure 3, α > 0). However, antipersistent fluctuations tend to systematically decluster recurrences, and the result is likewise a relatively uniformly distributed recurrence plot (Figure 3, α = −1).

Figure 4 shows the model coefficients for each α (*n* = 100 simulations each). This method seems appropriate for distinguishing among persistent signals (α > 0) for the idealized, but also noise data. It does not work for antipersistent fluctuations (α < 0). Here, the method simply does not distinguish between α-values of 0 and −1. For the noisy time series, we fitted linear regressions between α values and the power-law coefficients separately for the α ≥ 0 (*R*^2^ = 0.9) and α ≤ 0 (*R*^2^ = 0.04), which support the above statement.

#### 2.1.3. Second Approach: Estimating Scaling Using Laminarity *(%LAM)*

For this approach, we simply calculated the percentage of recurrence points that have a vertical/horizontal neighbor (*%LAM*, laminarity; [33]) over the whole plot (Figure 5). The rationale behind this approach is somewhat similar to the first approach, which is that fractal fluctuations tend to be manifested by patches or squares in the recurrence plot (see Figure 3). Hence, *%LAM* would represent the persistence of the data well.

The results corroborate this: persistent fractal fluctuations lead to increased laminarity with and without noise (Figure 6). In addition, there was a tight connection between the *%LAM* values and the true α-values, marked by a high R^2^ (0.96) quantifying correlation between α and *%LAM* for the noise condition.

While there is a mathematical relation between *%LAM* and autocorrelations in a time series, the method has a downside in that it does not capture scaling relations within the data per se, and hence represents more of a correlate of fractal fluctuations, albeit a very useful one.

#### 2.1.4. Third Approach: Estimating Scaling Relations via Diagonal Recurrence Rates *(Diag %REC)*

The third approach is based on diagonal recurrence profiles of a time series. The diagonal recurrence profile quantifies the number of recurrences at different lags, similar to the autocorrelation function [34]. To obtain the diagonal recurrence profile, one simply counts the proportion of recurrence points in the off-diagonals towards the lower-right or lower-left of the recurrence plot and plots them as a function of distance from the main diagonal; that is, lag [35]. Figure 7 illustrates the computation of the diagonal recurrence profile.

The rationale behind the approach is that the diagonal recurrence profile is a model-free type of autocorrelation [33,36], and hence captures the magnitude of autocorrelation at different lags, which is related to fractal fluctuations in a time series [37]. Accordingly, a scaling relation between the logarithm of the recurrence rate and the logarithm of the diagonal number (reflecting the frequency spectra) should be related to fractal scaling. Here, a sharper negative slope indicates dominance of lower frequencies. Hence, contrasting the previous approaches, a lower power-law coefficient evidence a more persistent fluctuation. Correspondingly to spectral scaling analysis, this method yielded a scaling exponent of 0 to white noise (α = 0)—a benchmark to determine whether the time series is persistent, random, or antipersistent.

As can be seen in Figure 8, this approach distinguishes comparatively well between the different exponents for persistent fluctuations, with and without noise, but is less sensitive to the antipersistent fluctuations (however, the exponents are still increasing with decreasing negative alpha-values). Moreover, the relation to the true α-values appears strong for this range, even with noise (*R*^2^ = 0.88). 

Another version of this approach is derived from Zbilut and Marwan’s [38] proposal, which applied the Wiener–Khinchin theorem [38] to the analysis of diagonal recurrence profiles. They show that one can detect (nonlinear) periodicities by applying a Fourier transform to the diagonal recurrence profile of an RP (Figure 7). Just as with the raw diagonal recurrence profile, we fitted a linear trend line to the log–log plot power spectrum (obtained via the Fourier Transform) of the diagonal recurrence profile (Figure 9). The results were similar to what we observed for the raw diagonal recurrence profile in that the method distinguished between persistent (*R*^2^ = 0.68, α ≥ 0) fluctuations. However, the standard errors were higher, and the method did not capture antipersistent fluctuations (*R*^2^ = 0.002, α ≤ 0).

#### 2.1.5. Fourth Approach: Consecutive Diagonals Recurrence Ratio *(Diag ratio)*

Until this point, the analysis techniques were more effective for persistence signals and did not distinguish between antipersistent signals well. Approach number four solves this issue to some degree. Similar to the third approach, we utilized the recurrence percentage of the diagonal lines. Here, however, we calculate the ratio between each couple of consecutive diagonal lines (Figure 7). The rationale behind the approach is that antipersistent fluctuations will tend to yield oscillations at high frequencies, and the ratio of recurrence rate of adjacent diagonals in the recurrence plot will capture the magnitude of such oscillations. Just as with the laminarity measure, however, this method is more of a correlate of antipersistent fractal scaling, and does not capture scaling properties directly.

As seen in Figure 10, with this measure, we can differentiate negative α-values (antipersistent) from α = 0, both with and without external noise. However, the method does not distinguish between the different alpha values of the persistent fluctuations.

### 2.2. Empirical Example

The approaches were tested on a dataset of a tapping experiment during which participants listened to a certain beat and were then instructed to tap according to the tempo they had heard. Under one of the two within-participant conditions, participants received visual feedback on every trial to help them align their tapping performance with the target tempo, while in the other condition no such feedback was provided. The sample was comprised of 36 time series from 18 participants with at about 1000 tapping intervals per time series.

Drawing on previous research on cognitive processes, we expected the time series to show persistent fractal fluctuation. Moreover, previous research showed that receiving feedback would reduce long-range dependencies in the data related to cognitive-motor processes of timing, and hence yield a more random (‘whiter’) noise manifested by a lower α exponent [39]. Our findings, displayed in Figure 11, support these expectations in several ways. Firstly, a negative power-law coefficient in approach 3 along with a ~1 ratio between subsequent diagonals (approach 4) indicate a persistent fluctuation in both conditions and is supported by a Hurst exponent 1.0 > *H* > 0.5, suggesting a pinkish noise. Further, approaches 1–3, as well as Wiener–Khinchin theorem’s results, imply a lower α-exponent for the feedback condition (see Table 1). While *SD %REC* and *%LAM* exhibit it by presenting a higher clustering characteristic for the no-feedback condition, *Diag %REC* and the Wiener–Khinchin theorem display it with a stronger lower frequency dominance when no feedback is given.

### 2.3. Comparison of the Approaches

To evaluate the presented approaches in relation to the true alpha values of the generated time series, we focus on three main parameters: (a) fractal dimension range, (b) sensitivity to noise, and (c) summary of the quantitative relation to the true alpha values. Furthermore, we investigated their applicability to empirical behavioral data.

#### 2.3.1. Range

As presented above, approaches 1 (*SD %REC*) and the Wiener–Khinchin-based analysis are sensitive to persistent fluctuations. Conversely, approach 4 (*Diag ratio*) differentiates only antipersistent fluctuations, whereas approaches 2 and 3 (*%LAM*, *Diag %REC*) are applicable throughout the whole tested range (−1 < α < 2), like DFA. Hence, with no estimation of the time series’ fractal dimension, one should conduct an analysis according to approaches 2 or 3, otherwise the researcher might prefer to pick the analysis technique that best fits his data’s characteristics. On a similar note, one can try to detect whether there are persistent fluctuations using approach 4, which yields a ~1 ratio for α ≥ 0.

#### 2.3.2. Robustness to Noise

Most of the analysis techniques that were applied were robust to noise. Except for the Wiener–Khinchin theorem approach, the rest distinguished between α-values within their range comparably with and without noise. Nevertheless, antipersistent fluctuations were less distinguishable by both approach 3 and DFA when i.i.d. noise (SNR = 2:1) was introduced.

#### 2.3.3. Quantitative Relation to True Alpha Values

Table 2 provides a summary of the *R*^2^-values that capture the relation between the true α-values and the estimated parameters of the different approaches, separately for persistent and antipersistent fluctuations. As DFA is the gold standard for fractal analyses in time-series methods, the comparison of the recurrence-based approaches to DFA is of particular interest here. Comparing the likelihoods of the linear models of each of our four approaches to DFA, we found that the association between the true α values and Hurst exponent is significantly stronger than in almost every other method (α < 0.05). On the contrary, approaches 2 and 4 yielded significantly higher association (than DFA) with the true α values for antipersistent fluctuation when noise is introduced, but somewhat below DFA under the no-noise condition. Nevertheless, approaches 1–3 show similar *R*^2^ to DFA when analyzing persistent noise. It has to be kept in mind that the sample sizes here are quite large, and tests of significance are of limited value in this case.

#### 2.3.4. Applicability

All approaches were found applicable to behavioral data and concluded conformally despite the small sample size. The utilized data were most likely to behave persistently and hence were out of the fourth approach range. Yet, we suggest using approach 4 to confirm whether the time series is persistent or not (persistent fluctuations are indicated by a 1:1 ratio between subsequent diagonals). Table 3 provides an overview of the *R^2^* for the different approaches (including DFA) when comparing the two time-estimation groups (i.e., with and without feedback). In a model comparison, approaches 2–4 were less predictive than DFA (α < 0.05), while approach 1 was not significantly lower.

## 3. Conclusions

In the current paper, we presented and compared several recurrence-based approaches to quantify the strength of monofractal autocorrelations in time-series data. This is a major step forward for integrating the quantification of scaling properties into recurrence quantification analysis, as previous research has suggested that such analyses are theoretically possible (e.g., [28]), but did not point to concrete means for how to deduce such properties. The proposed methods differ in quality, as well as in the range of applicability to particular types of colored noise, as we have shown on synthetic and empirical data. Based on our results, we recommend using approaches 3 and 4 to determine whether the data are persistent, antipersistent, or white noise. Then, approaches 1, 2, and 3 would be suitable to compare the fractal dimensionality of persistent data, while approaches 2 and 4 would fit antipersistent time series.

Thus, the present work lays the foundations for integrating fractal analysis into an RQA framework, and defining appropriate recurrence-based quantifies. Moreover, these methods might be amenable to quantifying time-dependent fractal fluctuations of not only univariate time series, but also strange attractor profiles, which possess fractal properties and are readily analyzable within the framework of recurrence quantification analysis [28]. In the future, these methods could be extended to capturing fractal dimensions in multidimensional systems via multidimensional recurrence quantification analysis. In addition, an evaluation and adaptation of these approaches to multifractals would be valuable [40,41].

## Figures and Tables

**Figure 1 entropy-24-01314-f001:**
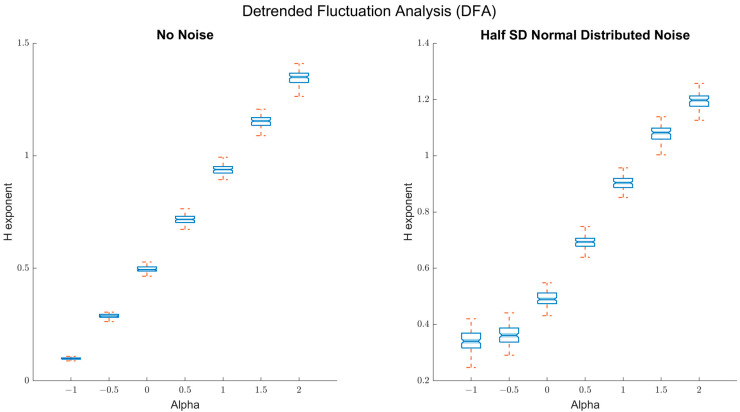
Detrended fluctuation analysis (DFA) results: Left panel: Box plots of the true α-values on the *x*-axis and the estimated Hurst exponents *H* on the *y*-axis from DFA. As can be seen, the DFA *H* scales well with the true alpha values down to antipersistent fluctuations (α = −1). Right panel: Box plots of the true α-values on the *x*-axis and the estimated Hurst exponents *H* on the *y*-axis from DFA, when random noise is added (SNR = 2:1). DFA still scales well for persistent fluctuations with the true α-values, but is relatively less sensitive to distinguishing between different types of antipersistent fluctuations.

**Figure 2 entropy-24-01314-f002:**
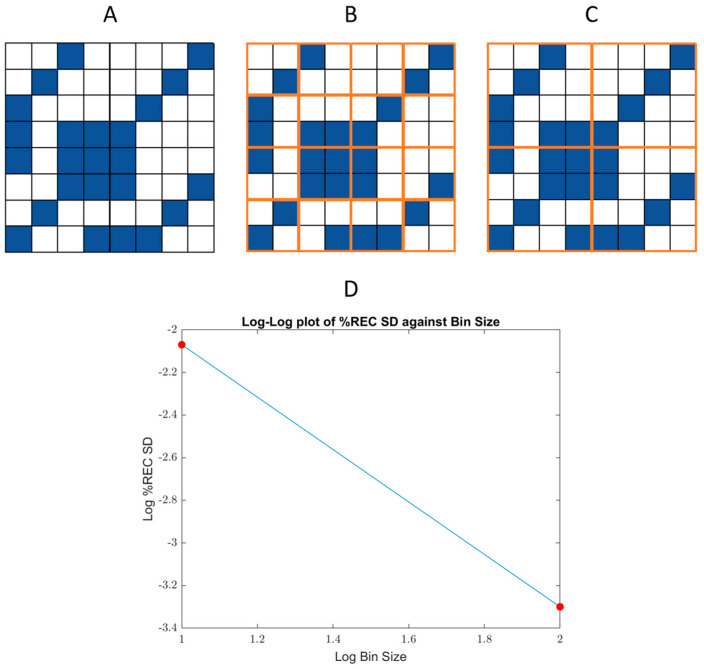
Demonstration of approach 1 over a simple recurrence plot: (**A**) A hypothetic 8 × 8 recurrence plot (RP) where blue squares stand for recurrence points and blank squares for nonrecurrent ones. In (**B**,**C**) the RP is split into bins of 2 and 4 (respectively, marked in brown). With approach 1, one finds the *%REC* in every bin and computes the *SD* between the recurrence percentages. Afterward, a linear trend is fitted to the log–log scaling plot (**D**) and the slope represents the scaling.

**Figure 3 entropy-24-01314-f003:**
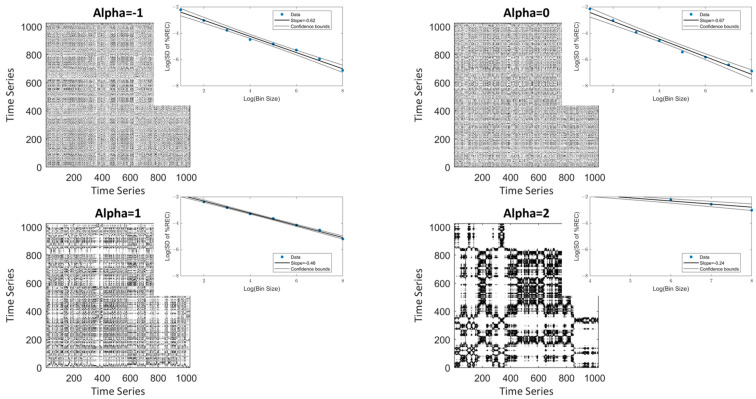
RP and scaling plots for different alpha values: Examples of univariate RP time series generated with different α-values, and scaling plots demonstrate the association of bin sizes and the *SD* of the recurrence rates between bins. As can be seen, from α = 0 the slope tends to decrease, suggesting a lower fractal dimension (i.e., higher α).

**Figure 4 entropy-24-01314-f004:**
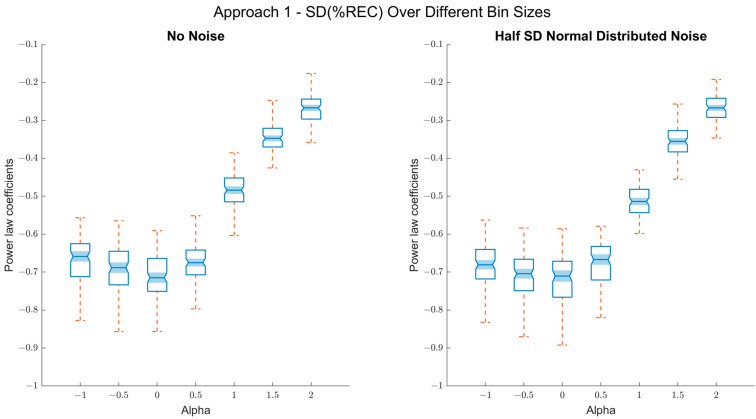
Results of approach 1 (*SD %REC*): Left panel: Box plots of the true alpha values on the *x*-axis and the power-law coefficients for the association of *SD* of *%REC* between the bins and bin sizes on the *y*-axis. As observed, the coefficient scales well with the true alpha values for the persistent fluctuations (α > 0). Right panel: Box plots of the true alpha values on the *x*-axis and the power-law coefficients for the association of *SD* of *%REC* between the bins and bin sizes on the *y*-axis, when random noise (SNR 2:1) is added. Still, the resulted coefficients scale well for persistent fluctuations with the true alpha values but are relatively insensitive to distinguishing between different types of antipersistent fluctuations.

**Figure 5 entropy-24-01314-f005:**
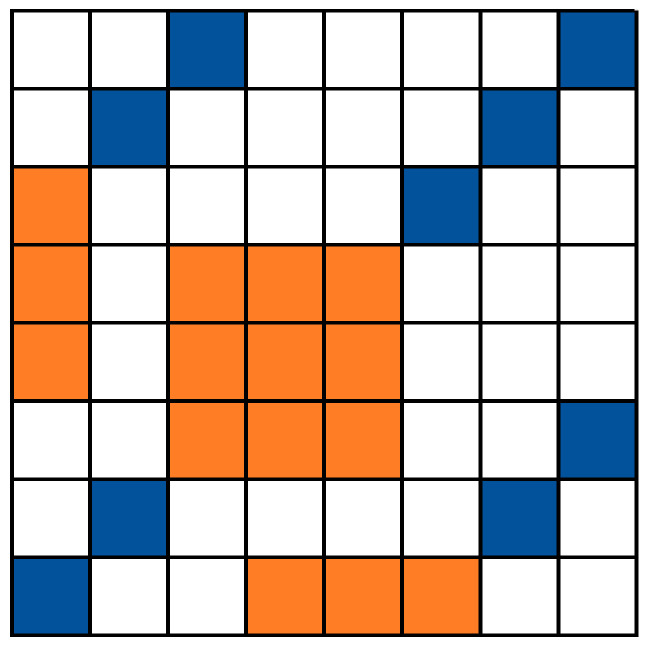
Quantifying laminarity: An 8 × 8 RP where colored squares represent recurrence points. The orange-filled recurrence points have a vertical/horizontal neighbor, while the blue squares do not. *%LAM* is the percentage of the recurrence points that have a vertical neighbor (orange) out of all the recurrence points (colored).

**Figure 6 entropy-24-01314-f006:**
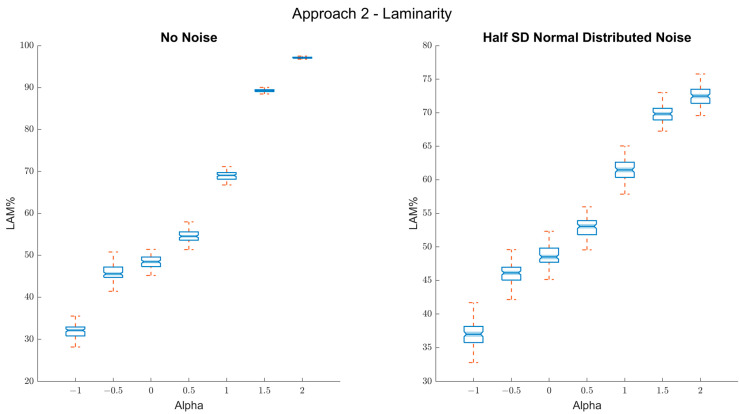
Results of approach 2 (laminarity). Left panel: box plots of the true alpha values on the *x*-axis and the *%LAM* on the *y*-axis. As can be seen, the coefficient scales well with the true α-values for both persistent and antipersistent fluctuations (α > 0). Right panel: box plots of the true α-values on the *x*-axis and the *%LAM* on the *y*-axis, when random noise (SNR 2:1) is added. Still, the resulted coefficients scale well for every alpha value (−1 < α < 2).

**Figure 7 entropy-24-01314-f007:**
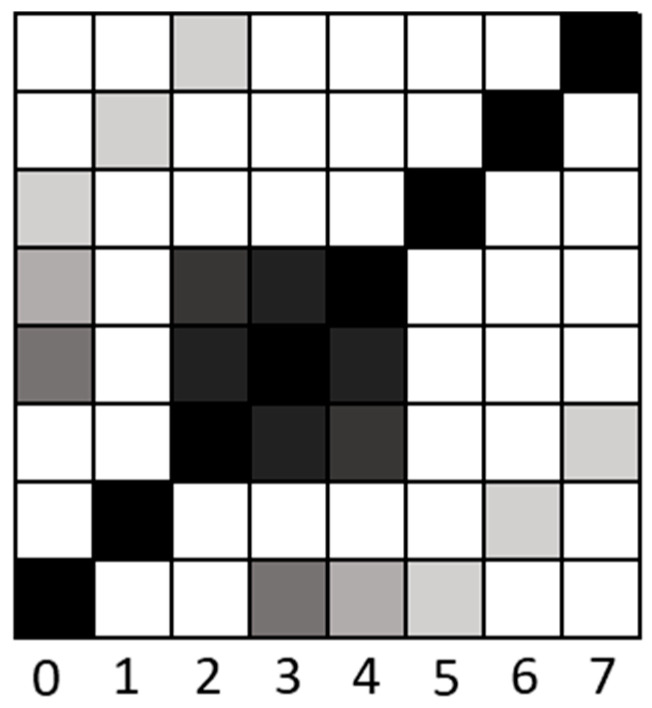
Approaches 3 and 4—diagonals in RP: An 8 × 8 RP presents the diagonal lines from 0 (main diagonal) to 7; due to the univariate RP’s symmetrical characteristic, only the bottom triangle was used. In approach 3, we counted the recurrence points in each diagonal and divided them by the diagonal’s length. Additionally, approach 4 utilizes the ratio of recurrence percentage between every two subsequent diagonal lines. Both approaches focused on the middle diagonals to avoid the main diagonal’s 100% recurrence points and the short diagonals towards the edges of the recurrence plot.

**Figure 8 entropy-24-01314-f008:**
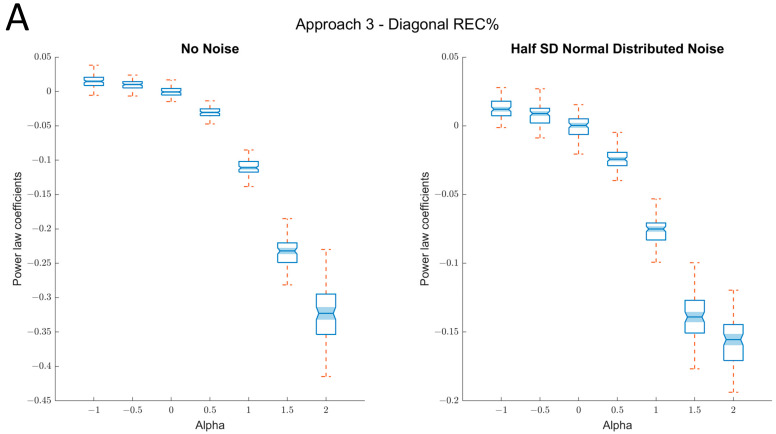
Results of approach 3 (*Diag %REC*). (**A**) Left panel: box plots of the true alpha values on the *x*-axis and the power-law coefficients for the association of diagonal *%REC* and the diagonal index (distance from the main diagonal) on the *y*-axis. As can be seen, the coefficient scales well with the true α-values for both persistent and antipersistent fluctuations (−1 < α < 2). Right panel: box plots of the true alpha values on the *x*-axis and the power-law coefficients for the association of diagonal *%REC* and the diagonal index on the *y*-axis when random noise (SNR = 2:1) is added. Still, the resulted coefficients scale well with the true α-values for persistent and antipersistent fluctuations, but are somewhat less sensitive to distinguishing between different types of antipersistent fluctuations (α < 0). (**B**) Example of scaling plots demonstrating the association of diagonal *%REC* and diagonal index for different α values.

**Figure 9 entropy-24-01314-f009:**
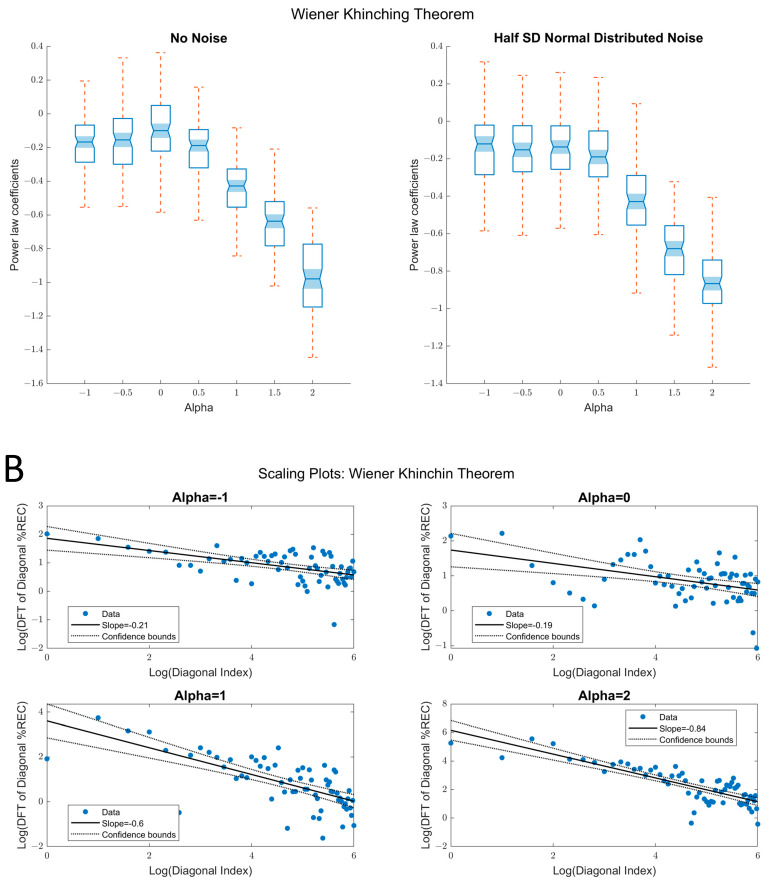
Results of Wiener–Khinchin theorem. (**A**) Left panel: box plots of the true α-values on the *x*-axis and the power-law coefficients for the association of FT of the diagonal *%REC* and the diagonal index on the *y*-axis. As can be seen, the coefficient scales well with the true α-values for the persistent fluctuations (α > 0). Right panel: box plots of the true α-values on the *x*-axis and the power-law coefficients for the association of FT of the diagonal *%REC* and the diagonal index on the *y*-axis, when random noise (SNR = 2:1) is added. The relation of the resulting coefficients to the true α-values is not as good for persistent fluctuations (cannot differentiate α = 0 and 0.5) and is relatively insensitive to distinguishing between different types of antipersistent fluctuations. (**B**) Example of scaling plots demonstrating the association of FT of the diagonal *%REC* and diagonal index for different α values.

**Figure 10 entropy-24-01314-f010:**
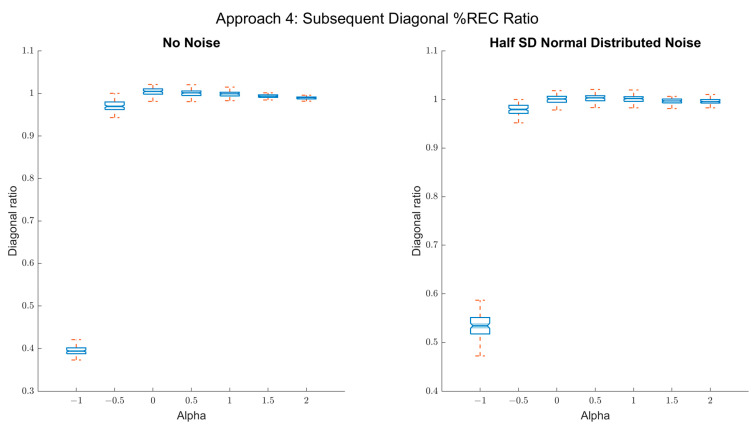
Results of approach 4 (consecutive diagonals *%REC* ratio). Left panel: box plots of the true alpha values on the *x*-axis and the mean ratio between subsequent diagonals’ *%REC* on the *y*-axis. As can be seen, the coefficient scales well with the true α-values for antipersistent fluctuations (α < 0) and converges to 1 from α = 0. Right panel: box plots of the true α-values on the *x*-axis and the mean ratio between subsequent diagonals’ *%REC* on the *y*-axis when random noise (SNR 2:1) is added. Still, the resulted coefficients scale well for negative alpha value (*R*^2^ = 0.79).

**Figure 11 entropy-24-01314-f011:**
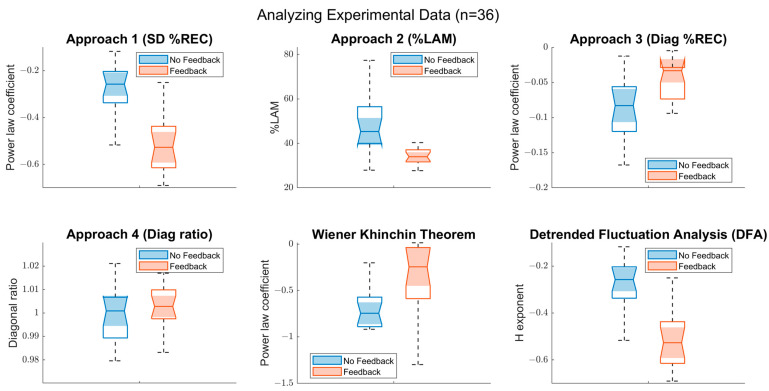
Box plots illustrating the outcomes of feedback and no-feedback conditions: Six sets of box plots represent a comparison between the outcomes of each approach for feedback (orange) and no-feedback (blue) conditions. While the frame of the boxplot is defined by the interquartile range, the notch represents a 95% confidence interval and the whiskers show the maximum and the minimum of each distribution (except outliers). As expected, due to its persistent noise characteristics (α > 0), behavioral data would be appropriately analyzed by approaches 1–3 but not approach 4. Approaches 1 and 2, as well as DFA, yield higher results for the no-feedback condition, indicating a larger α, meaning a more persistent behavior. Likewise, approach 3 and Wiener–Khinchin theorem suggest a lower frequency dominancy in the no-feedback condition.

**Table 1 entropy-24-01314-t001:** Paired *t*-test results comparing the outcomes of feedback and no-feedback conditions.

Approach	*t*	*df*	*p*
1—*SD %REC*	−5.12	17	>0.001
2—*%LAM*	−3.33	17	0.004
3—*Diag %REC*	3.43	17	0.003
4—*Diag ratio*	0.82	17	0.42
Wiener–Khinchin theorem	3.68	17	0.002
DFA	−4.08	17	>0.001

**Table 2 entropy-24-01314-t002:** Comparison of approaches on simulated data.

Approach	*R* ^2^ *—Persistent (No Noise)*	*R* ^2^ *—Antipersistent (No Noise)*	*R* ^2^ *—Persistent (with Noise)*	*R* ^2^ *—Antipersistent (with Noise)*
1*—SD %REC*	0.9	0.08	0.9	0.04
2*—%LAM*	0.97	0.82	0.95	0.81
3*—Diag %REC*	0.93	0.38	0.93	0.33
4—*Diag ratio*	0.33	0.79	0.06	0.78
Wiener–Khinchin theorem	0.7	0.01	0.68	0.002
DFA	0.99	0.99	0.98	0.68

**Table 3 entropy-24-01314-t003:** Comparison of approaches on empirical data.

Approach	*R^2^**—with* vs. *without Feedback*
1*—SD %REC*	0.33
2*—%LAM*	0.11
3*—Diag %REC*	0.25
4—*Diag ratio*	0
Wiener–Khinchin theorem	0.23
DFA	0.36

## Data Availability

The data is available upon request from the corresponding author. Also, a Matlab function (The MathWorks, Inc.) to perform the four analyses is available on GitHub, see Appendix A.

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
