# Peer review of "Four Methods to Distinguish between Fractal Dimensions in Time Series through Recurrence Quantification Analysis"

_entropy, 2022, doi:10.3390/e24091314_

Round 1

Reviewer 1 Report

The authors of the paper describe their proposed approach for Four methods to distinguish between fractal dimensions in time series through Recurrence Quantification Analysis. The topic is interesting and with possible applicability. However, the paper needs several improvements:

1) the main contribution and originality should be explained in more detail, which part of the proposal is new?

2) the motivation of the approach with needs further clarification, why this work is needed or is important?

3) discussion of related work in fractal dimension for time series should be expanded with more recent work

4) Minor grammar and syntax issues need correction

5) more simulation results and formal comparison of results are needed

6) the conclusions should be extended with more future work

7) Transitions from section to section should be smoother

8) More references to fractal dimension for time series papers could be included, like for example, the following:

A novel method for a covid-19 classification of countries based on an intelligent fuzzy fractal approach

Forecasting of COVID-19 time series for countries in the world based on a hybrid approach combining the fractal dimension and fuzzy logic

Author Response

The authors of the paper describe their proposed approach for Four methods to distinguish between fractal dimensions in time series through Recurrence Quantification Analysis. The topic is interesting and with possible applicability. However, the paper needs several improvements:

Dear reviewer, thank you for your comments. Below you will find our answers to your suggestions.

1) the main contribution and originality should be explained in more detail, which part of the proposal is new?

We have elaborated on the aim of the paper, and have tried to clarify what the contribution is (mainly providing methods of estimated fractal scaling properties of time series using recurrence quantification analysis, which the contemporary methods do not do).

2) the motivation of the approach with needs further clarification, why this work is needed or is important?

We have tried to clarify this better in the introduction (see also our response to point 1 above).

3) discussion of related work in fractal dimension for time series should be expanded with more recent work

We have added more references, particularly using DFA (see also our response to your comment 8). However, we are not aware of any related work that links recurrence analysis to fractal analysis, except the paper by Charles Webber that we cite in the paper (Webber, 2012).

4) Minor grammar and syntax issues need correction.

We have read through the whole paper to improve the language.

5) more simulation results and formal comparison of results are needed

Added to comparison. We tested the methods with association to the true alpha values.

6) the conclusions should be extended with more future work

We have added more on future work, particularly on mdRQA as well as multi-fractals.

7) Transitions from section to section should be smoother.

We have read through the whole manuscript to improve the language and the flow of the sections.

8) More references to fractal dimension for time series papers could be included, like for example, the following:

We have added several newer references to highlight recent work, particularly also using DFA, which is the gold standard to capture fractal scaling in time series data.

Reviewer 2 Report

Improve the introduction without falling, for example, into the use of ambiguities such as "Fractal 14 properties are marked (imperfect) recurrences..." Eliminate what is in parentheses.

Extend the conclusions in such a way that they reflect all the simulation work carried out.

Author Response

Dear reviewer, thank you for your comments. Below you’ll find our answers to your suggestions.

Improve the introduction without falling, for example, into the use of ambiguities such as "Fractal 14 properties are marked (imperfect) recurrences..." Eliminate what is in parentheses.

We have rephrased the sentences and read through the whole paper again to eliminate ambiguities.

Extend the conclusions in such a way that they reflect all the simulation work carried out.

We added some recommendations according to the results. Moreover, we added tables and quantified the performance of the methods in the conclusion section.

Reviewer 3 Report

This manuscript presents the four methods for calculating FD based on RQA. This is an exciting topic; however, some aspects should be improved. For this purpose, I suggest the following recommendations:

1.      The novelty of this paper is not clear. More details are needed.

2. English could be improved since it is strange and incomprehensible in some places.

3.      The analysis of the simulation results is weak. How can the obtained results be interpreted?

4.   The references are not new. Some recently published papers should be inserted.

5. Compare the obtained results with the other existing methods. Comparison between the proposed methods is necessary, however it is not sufficient.

Author Response

This manuscript presents the four methods for calculating FD based on RQA. This is an exciting topic; however, some aspects should be improved. For this purpose, I suggest the following recommendations:

Dear reviewer, thank you for your comments. Below you’ll find our answers to your suggestions.

  1. The novelty of this paper is not clear. More details are needed.

We have elaborated on the aim of the paper, and have tried to clarify what the contribution is (mainly providing methods of estimated fractal scaling properties of time series using recurrence quantification analysis, which current methods do not do).

  1. English could be improved since it is strange and incomprehensible in some places.

We have read through the whole paper again and improved the language.

  1. The analysis of the simulation results is weak. How can the obtained results be interpreted?

We added some recommendations according to the results. Moreover, we added tables and quantified the performance of the methods in the conclusion section.

  1.  The references are not new. Some recently published papers should be inserted.

We have added several newer references to highlight recent work, particularly also using DFA, which is the gold standard to capture fractal scaling in time series data.

  1. Compare the obtained results with the other existing methods. Comparison between the proposed methods is necessary, however it is not sufficient.

Our methods were compared both to the predefined alpha value of the generated time series and to the widely approved DFA method. Now, we have added a summary of the methods’ performance and a comparison to DFA.